# Companion Animal Ownership and Human Well-Being in a Metropolis—The Case of Hong Kong

**DOI:** 10.3390/ijerph16101729

**Published:** 2019-05-16

**Authors:** Paul W.C. Wong, Rose W.M. Yu, Joe T.K. Ngai

**Affiliations:** Department of Social Work and Social Administration, The University of Hong Kong, Pokfulam, Hong Kong, China; yuwaiman@hku.hk (R.W.M.Y.); joe.tkngai@hku.hk (J.T.K.N.)

**Keywords:** human–animal interactions, companion animal ownership, human well-being, Hong Kong

## Abstract

Global urbanization has given cause for a re-assessment of the nature and importance of the relationship between humans and domesticated animals. In densely-populated urban societies, where loneliness and alienation can be prevalent, the use of animals as human companions has taken on heightened importance. Hong Kong is the world’s most urbanised political entity, and thus provides an ideal context for the exploration of the role of animals in the provision of companionship for human beings in cities. A web-based survey with descriptive analyses, regression, and ANOVA was conducted. Six-hundred-and-forty-seven companion animal owners and 312 non-owners completed the survey that examined their socio-demographic information, companion animal ownership status, and physical-psychosocial well-being. The statistically significant findings appear to suggest that socio-demographic variables (i.e., age, gender, housing, and education level) have stronger predictive values than companion animal ownership status with respect to the well-being of people in Hong Kong. Due the unique environmental features in Hong Kong, the positive impacts of companion animal ownership on the physical well-being of owners may be limited by the city’s cramped living space and the limited number of people who own companion animals. However, results suggested that companion animals may still serve as a social lubricant between the owners and their significant others, thereby playing a heightened role significant role in enhancing general social connectedness in a metropolis. Given the importance of animals as human companions, it is suggested that relevant administrative agencies need to consider the development of policies and facilities which are conducive to both the maintenance and development of the bonds between humans and their companion animals and the physical and psychosocial health of both.

## 1. Introduction

Historically, humans have domesticated animals mainly for their instrumental value (that is, as property, food, and the provision of labour and fibres) [1]. At the same time, there is a long history of the use of animals as companions. The Ancient Egyptians had a penchant for cats, monkeys, and falcons. The Ancient Greeks kept dogs as household pets [2]. Certainly, over the course of history, animals have become increasingly important as companions for humans. In many cases, animal companionship has become a human need, never more so than now in large urban centres.

The urban population of the world has grown rapidly, from 751 million in 1950 to 4.2 billion in 2018, and by 2050, 68% of the world’s population will live in urban areas [3]. Such a major population shift has significant consequences on our well-being. In particular, urbanization seems to have contributed to the reduction in marriage rates and increased rates of childlessness, resulting in smaller household sizes. Consequently, for many people, the breadth of their social networks has been reduced both in terms of their size and their capacity for intimacy [4]. These demographic changes seem to have contributed to increased rates of social isolation and loneliness among both young [5] and older people [6]. A further outcome of these societal changes has been the development of a more sedentary lifestyle [7], a state of affairs which can have adverse effects on both the physical and psychological well-being of individuals.

As a result of the increasing rate of global urbanization, in particular the spread of mega-cities in the last decades, the role of animals as companions in enhancing human physical and psychosocial well-being has become a shared interest among animal welfare advocates, animal owners, and the individuals and organisations whose areas of interest and expertise lie within this field of study [8]. Whilst acknowledging that studies on the interaction between humans and companion animals are limited, what has emerged from the studies is a general support for the notion that owning a companion animal has positive influences on the physical, psychological, and social well-being of human beings [9]. Physically, ownership of companion animals has been associated with higher survival rates from cardiovascular diseases [10,11], lower cholesterol and blood pressure levels [12,13], fewer reports of minor health problems such as headaches, colds, and dizziness [14], and fewer doctors’ consultation sessions [15]. In the long term, animal companionship has the propensity to reduce the community’s healthcare costs [14,16]. Psychosocially, ownership of companion animals is associated with a higher level of self-esteem and the development of autonomy in children. It is claimed that it is a protective factor against the onset of loneliness among the elderly [17,18,19]. It can result in the reduction of anxiety, the adoption of a more positive outlook on life, and a greater perceived competence on the part of senior members in society [20].

Despite this general consensus with respect to the benefits of owning or having access to a companion animal, it needs to be taken into account that the relationship between companion animal ownership and the enhanced well-being of individuals is a complex phenomenon and can be influenced by a significant number of factors, all of which can cause variance in the nature and extent of these benefits. In the literature published to date on this issue, the gender, age, marital and socio-economic statuses, type of dwelling, and availability of social support of the owners have been identified as common confounders [21], challenging prevailing beliefs about the value and nature of ownership of companion animals. For instance, in a cross-sectional web-based survey on the association between companion animals and depression, it was found that principally, marital status and gender were the factors which affected the depression score of the participants. Furthermore, the protective effect of having either a dog or a cat was only valid for unmarried women. In fact, unmarried men with a dog or cat were reported to have the highest level of depression symptoms [22]. Again, an increase in age on the part of the owner of a companion animal has also been found to be negatively associated with the owner’s well-being, possibly due to their diminishing ability to take care of the animal and their fear of loss and separation [23,24]. Companion animal owners who have high level of social support have been found to benefit more from animal ownership than those with a lower level of support [25].

Hong Kong is one of the world’s most populous and urbanized cities. Almost 100% of the population in Hong Kong live in developed areas [3], with a population density of 6830 people per square kilometre, with most living in high-rise buildings [26]. Consequently, the median living space per capita in Hong Kong is 1.6 square metres, amongst the smallest in the world [27]. Hong Kong is thus one of the most suitable “urbanized population laboratories”, in which to study human–animal interactions (HAI). It is hoped that this investigation will provide insights into the nature of animal companionship and those conditions, under which individuals living in densely-populated urban environments can gain maximum benefit from their ownership or access to companion animals. This study had three aims: (1) to compare the state of well-being between companion animal owners and non-owners in a metropolis; (2) to identify specific socio-demographic factors that may influence the viability and efficacy of companion animal ownership; and (3) given that the majority of the literature focused on dog ownership and well-being, it aimed to identify if there were differences between dog ownership and other non-canine animal ownership on the well-being of people in Hong Kong.

## 2. Materials and Methods

### 2.1. Methodology

The study was conducted between 20 June and 19 July 2016. Participants were recruited through the issue of study invitation messages disseminated through the websites and Facebook pages of animal welfare organizations, organizations providing animal-assisted interventions, a local youth organization, and through the network of the organisations and interested individuals. Informed consent was obtained prior to the commencement of the bilingual (Chinese and English) 30-minute, web-based survey. Incentives in the form of goods coupons (around US$8) were offered on completion of the survey. Measures used in the questionnaire that had no equivalent in the Chinese language were back-translated by two bilingual mental health professionals. A pilot test with 10 individuals, including both owners and non-owners of companion animals, was conducted in order to assess the readability of the questionnaire. This study was approved by the Human Research Ethics Committee for Non-Clinical Faculties at the University of Hong Kong (Reference number: EA1604001).

### 2.2. Measurements

The questionnaire had five domains: (1) socio-demographic information; (2) physical well-being; (3) psychological well-being; (4) social well-being; and (5) companion animal ownership status.

#### 2.2.1. Sociodemographic Variables

The socio-demographic variables measured in the study were gender, age, education level, household size, household income, and type of housing.

#### 2.2.2. Physical Well-Being

Subjective health: subjective perception of health was considered a valid proxy measure to assess the health status of an individual. Hence, the self-report single-item question “the number of days he/she rated his/her physical health as ‘not good’ over the past 30 days” from the Health-related Quality of Life (HRQOL) was adopted and the number was then constituted as the score [28].

Physical activities: the health benefits of regular walking have been well documented [29,30] and it has been found that dog owners were more likely to engage in walking [31]. A question that asked the participants to state how frequently they walked for at least 10 minutes at a time during the course of a week was adopted from a health-related behaviour survey conducted by the Department of Health of the Hong Kong Special Administrative Region of the People’s Republic of China (HKSAR) Government [32].

#### 2.2.3. Psychological Well-Being

The Depression, Anxiety and Stress Scale (DASS-21) is a self-report 4-point Likert scale designed to measure, over the last seven days, symptoms of depression, anxiety, and stress [33]. The Chinese version of the DASS-21 [34] was adopted. In this study, the Cronbach’s alpha ranged from 0.86 to 0.90, indicating a high level of reliability.

#### 2.2.4. Social Well-Being

The Basic Needs Satisfaction in Relationship Scale (BNS-R) was developed based on the assumption that the needs for competence, autonomy, and relatedness are the three basic psychological needs of an individual [35,36,37]. This scale consists of 15 items describing an individual’s experience when relating to significant others. The participants were asked to indicate how accurately the statement reflected their feelings on a 7-point Likert scale. In this study, “significant other” was defined as a spouse and/or best friend. The reliability of BNS-R is 0.92, indicating a high degree of reliability.

#### 2.2.5. Companion Animal Ownership Status

Companion animal ownership status was first examined by asking the participants whether they were current owners, and, if yes, they were asked what type of animal(s) they kept. The classification of types of animals, namely dogs, cats, turtles/tortoises, birds, hamsters, and rabbits, was adopted based on the two household surveys conducted by the Hong Kong SAR Government [38,39].

### 2.3. Statistical Analysis

A three-stage statistical analysis was conducted by the IBM SPSS statistics version 23 (International Business Machines Corporation, New York, NY, USA). Firstly, descriptive statistics were conducted to examine the socio-demographic profiles of the participants. Secondly, Poisson regression analysis and two-ways ANOVA were conducted to explore the interrelationships between the socio-demographics, companion animal ownership status and the physical, psychological and social wellbeing of the participants. Thirdly, a sub-group analysis using simple descriptive statistics was conducted to compare the well-being of dog owners and other animal owners without dogs.

## 3. Results

### 3.1. Participants

A total of 986 participants completed the survey. Table 1 shows the detail of the profiles of the participants. Among them, 68% (*n* = 674) reported that they had one or more companion animal(s) in the household. The majority of participants were female (80%), aged between 20 to 39 (58%), had attained at least college education (67%) and had a household income of more than HK$40,000 (equivalent to around US$5000). More than half of the participants were living in non-subsidized (private) housing (58%), with two or more persons in the household (89%). In line with global trends, dogs were the most popular companion animals. Among owners, 72% kept at least one dog (*n* = 483; 49%) and 30% kept at least one cat (*n* = 204). The next popular type of companion animal was turtle (*n* = 80, 12%).

In comparisons between owners and non-owners, statistically significant differences were found, in terms of gender, age, income level, education, type of residence and number of occupants in the household. Owners were more likely to be female (85% among owners and 68% among non-owners), be aged between 40 and 59 (40% among owners and 24% among non-owners), and have a higher income level (>HK$40,000 per month) (47% among owners and 38% among non-owners).

Housing type was also a significant factor affecting companion animal ownership. Participants residing in public and subsidised housing were significantly more likely to be non-owners (24% owners and 41% non-owners), in all likelihood because of the dog prohibition rule in public rental housing flats in Hong Kong. In terms of number of occupants in the household, owners were more likely to be living alone or in two-member households (13% single and 39% two-member households were owners; 6% single and 24% two-member households were non-owners). As the number of occupants increased, the likelihood of being an owner was significantly reduced (30% of three-member households and 40% of >4-member households were non-owners; 21% of three-member households and 27% of >4-member households were owners).

### 3.2. Comparison of Physical and Psychosocial Well-Being between Owners and Non-Owners 

The differences in the physical and psychosocial wellbeing between owners and non-owners were examined using an independent sample *t*-test. Contrary to the expectation that companion animal owners were generally healthier, the mean number of unhealthy days in the past 30 days was reported to be 2.99 days by animal owners, and just 1.97 for non-owners. On the other hand, owners reported that they were more likely to walk for 10 minutes or more per day. With respect to psychological well-being, companion animal owners had a marginally lower level of anxiety, in comparison to non-owners; however, statistically significant differences in terms of the levels of depression and stress was not observed. Socially, owners reported a higher level of needs satisfaction in terms of autonomy, competence, and relatedness in their relationship with their significant others (see Table 2).

### 3.3. Moderation Effect of Companion Animal Ownership between Sociodemographic and Well-Being Variables

To explore the moderation effect of companion animal ownerships between sociodemographic and well-being variables, Poisson regression was conducted for physical well-being as the response variable that is a continuous variable. Physical well-being was represented as the number of unhealthy days in the past month. ANOVA was conducted for psychosocial well-being variables as they are categorical variables. Psychological well-being was represented by the levels of depression, anxiety, and stress. Social well-being was represented as the levels of needs satisfaction.

### 3.4. Physical Well-Being

#### 3.4.1. Number of Unhealthy Days

The result of the Poisson regression showed the relative ratio of the occurrence of the number of physically unhealthy days in relation to ownership status across age groups (see Table 3). Significant findings were identified for respondents under 19 years old and those aged between 19–39 years. Results suggested that owners in these two age groups were likely to experience more physically unhealthy days within 30 days prior to the completion of the survey. For the group aged under 19 years old, owners were 5.54 times more likely to experience physically unhealthy days than non-owners. In the case of those aged between 19 and 39, the relative ratio of owners experiencing an unhealthy day was 1.57 times more than non-owners. A similar result was also recorded among those aged between 40 and 59 years: owners were 1.17 times more likely than non-owners to experience ill health within 30 days prior to data collection. Nonetheless, results should be interpreted with caution as the age distribution in the sample was to be skewed. The number of individuals in the age group under 19-year group was limited and there were only 13 respondents in the age group ≥60.

#### 3.4.2. Frequency in Walking 10 Minutes or More

No statistically significant interactional effects were found for the sociodemographic variables, ownership status, and the level of physical activities in terms of frequency of walking 10 minutes or more.

### 3.5. Psychological Well-Being

No statistically significant interaction effect was found for the socio-demographic variables, ownership status and the psychological outcomes, in terms of depression, anxiety, and stress. Thus, in short, companion animal ownership has no statistically significant impact on the variations in the DASS21 scores reported by the participants from different socio-demographic profile.

### 3.6. Social Well-Being

The means and standard deviations of the BNS-R score by age group and ownership status were presented in Table 4. There were no statistically significant differences between owners and non-owners in different age groups. The results of the two-way ANOVA that examined the main and interaction effects of companion animal ownership and age on the BNS-R score were presented in Table 5. While controlling the effect of household income, results indicated that age did have significant impact on the score of BNS-R, while companion animal ownership did not. Older respondents seemed to experience a higher level of basic need satisfaction in their relationship with significant others. However, companion animal ownership tended to moderate the impact of age. Figure 1 outlined the mean difference of BNS-R between owners and non-owners by age. In general, the BNS-R mean score of owners was higher than that of their non-owner counterparts, with the exception of the age group of 60 and over. However, this should be interpreted with considerable caution as there was only one case in the ≥60 non-owner category.

In addition, statistical analysis also revealed that ownership status moderates the effect of education level on BNS-R score. Table 6 shows the means and standard deviations of the BNS-R score by ownership status across different education levels. The results of ANOVA displayed in Table 7 indicated that both education level and ownership have significant effects on the BNS-R score. Respondents with higher education level tended to obtain a higher BNS-R score. Moreover, a significant interaction effect between ownership status and education level was found in the analysis. Companion animal ownership moderates the effect of education on social outcome. Figure 2 illustrates the moderation effect of companion animal ownership. It indicated that companion animal owners have higher BNS-R score across different education levels; however, its impact was the most significant for those with primary or lower education level.

### 3.7. Comparisons of Physical, Psychological, and Social Well-Being between Dog Owners and Non-Dog Animal Owners

To examine whether dog ownership (including those who kept more than one type of companion animal beside a dog) may lead to similar physical and social well-being outcomes in Hong Kong when compared with non-dog owners, we conducted a simple subgroup analysis. Among the 674 companion animal owners, three participants did not specify the type of their companion animal and were excluded in the analysis. Among the 671 owners, 483 (72%) were dog owners and 187 (28%) were non-dog owners. No statistically significant difference was found between the number of unhealthy days reported by dog owners and non-dog owners. Although dog owners’ mean for unhealthy days was higher, the high value of the standard deviation suggested that there was a large variation within the group. In terms of psychological and social well-being, no statistically significant difference was found between the two groups, in terms of depression, anxiety, and stress, and their BNS-R scores (see Table 8).

## 4. Discussion

Examinations on the relationships between companion animal ownership and human well-being have been mainly conducted in countries such as Australia, Canada, Germany, New Zealand, United Kingdom, and the United States [14,15,21,24,40,41,42,43], where companion animal ownership is more common, finding that attitudes towards animals are generally positive and there is a more fluid spatial boundary between human and non-human animals [44,45]. Hong Kong provides a unique context in the study of HAI, especially as it is anticipated that globally more people will live in cities, especially mega-cities, in the coming decades. The exploratory nature of this study provides a comparative snapshot of the profiles of people with and without companion animals in terms of the effects that this state of affairs has on their physical and psychosocial well-being.

We freely acknowledge the limitations of this study. This study was the first of its kind in Hong Kong to capture a snapshot of the local situation of HAI between companion animal owners and their animals. One major factor that limits the generalisability of our findings is the lack of a representative sample, due to the unavailability of the amount of research funding that was needed to support this relatively small, but significant research theme in Hong Kong. We were restricted to a web-based study. Consequently, there is oversampling with respect to the number of participants who were companion animal owners. The uneven number of the participants in subgroup affects the validity of the result; for example, most of the participants were animal owners in the age range from 20 to 59. Again, limited by the scale of the study, we also had to strike a balance between the thoroughness of the question items and the estimated time for respondents to complete the questionnaire. We acknowledge that the adequacy of the single-item, subjective physical health measure may well be considered as an issue in terms of the validity of the conclusions drawn from the responses of participants. With more substantial researching interest and support in the future in Hong Kong, it is recommended that a more rigorous study with better sampling methods be conducted.

Nevertheless, the findings of this study when interpreted with caution, can serve as indicators for the future development of human-animal bonding and research activities ranging from animal adoption to animal-assisted activities in Hong Kong. Our tentative findings seem to suggest a need to rethink those published findings with respect to the value and nature of the relationships between human and companion animals. The findings seem to arrive at a different understanding with respect to what has been reported in current literature about human well-being and companion animal ownership. We are certainly not questioning these findings, but simply making the point that the relationship between humans and companion animals may well vary among individual and age groups, and that there are significant and diverse variables which impact on the nature and efficacy of such relationships. Previous studies have mainly highlighted the positive impacts of companionship of animals on human well-being through increased physical activities and social interactions with others; however, our findings tend to suggest that socio-demographic variables (i.e., age, gender, housing, and education level) have stronger predictive values than companion animal ownership status per se on the well-being of people in Hong Kong. The difference between the physical and psychological well-being of owners and non-owners would not seem to be as pronounced as previous researchers have hypothesised.

Our findings tend to concur with emerging research evidence which suggests that the influences of companion animal ownership on human well-being are complex, and that gazetted positive relationships can be mediated and moderated by a significant and diverse range of factors. For instance, one of the largest studies on human physical well-being and companion animal ownership, based on a population-based, random sampled of over 40,000 individuals residing in California, found that the general health ratings of dog and cat owners were slightly higher than of non-owners; however, this difference evaporated when factors like income, race, and marital status were taken into consideration. In short, when demographics and socioeconomic factors were controlled, researchers found no evidence that companion animal ownership was related to better health in the respondents [46].

Among the three principal domains of well-being that we examined, only the social well-being of owners seemed to have been positively influenced by companion animal ownership. This finding, however, may be better understood if viewed from the perspective of the unique environment of Hong Kong’s dense, urban ecology. In Hong Kong, dogs remain the main category of companion animal in the household. However, in such extremely limited living space, and a generally no-dog policy in many public areas, dog-walking could be less frequent and may be even stressful when compared to less urbanized societies. Dog walking in Hong Kong often creates arguments between dog owners and others (for example, who has the priority to use the lift in high-rise buildings, who has a say in setting the “standards” in cleaning after dog fouling, or whether the length of leashes creates a “safe” distance between non-owners and dogs). Hence, dog walking may not be perceived as positive or as an incentive to physical activity, as it would be in other places. Another perspective could be that given that public transportation is more accessible in urbanized metropolises, city people who reside in much smaller communities walk more than people who rely on driving their own vehicles to commute to places. In fact, according to a very large study based on a dataset covering 717,527 people using the smartphone app Argus, which tracks physical activity across 111 countries, it was very interesting to find that Hong Kong people walk the most (on average 6880 steps per day compared to the world average of 4961 steps) and with a very small range of activity inequality as compared to other studied countries [47]. In other words, Hong Kong people, in general, walk a lot already regardless of whether they are companion animal owners or not.

The combination of non-statically significant findings pertinent to the effects of companion animal ownership on physical and psychosocial well-being of the participants, as opposed to moderate statistically significant findings on social well-being in Hong Kong, is quite intriguing. We would argue that whereas there is a positive correlation between the ownership of a companion animal and the social well-being of individuals, unique ecological factors in Hong Kong may diminish the extent of this impact. In this study, we only examined the impact of companion animal ownership on their owners in relation to their significant others. It would be interesting to see whether the same effects would be observed on owners with others social relationships besides those with significant others (for example, neighbours).

## 5. Conclusions

Two aspects are certain. The first is that urbanisation continues to spread unabated. Today, some 54% of the world’s population lives in urban centres; by 2050, that percentage will rise to 65% [48]. Today, there are some 29 mega-cities (10 million plus population); six more cities, principally in Asia and Africa, will become mega-cities by 2030. The second certainty is that, with urbanisation, especially in the shape of mega-cities, there is an epidemic raging. This is not only related to chronic physical diseases like hypertension and obesity; there is also an epidemic of loneliness and alienation which is taking its toll on both young and old. This is poignantly illustrated by the fact that, in contemporary Tokyo for instance, socially withdrawn young people can rent a “sister” to visit them at home and talk to them, and some isolated elderly die alone with their bodies discovered months or even years later. Human beings were never meant to be solitary creatures. Consequently, the dearth of human connectedness has an adverse effect on human physical and mental well-being. This is a public health and an economic problem for society as a whole.

We are of the belief that animal companionship, if supported and nurtured by wider society and the government which represents it, is an important means of reducing loneliness and alienation of individuals of diverse ages and cultural and socio-economic backgrounds. As residents of Hong Kong, we are very much aware that there are any number of prevailing attitudes and norms which militate against an environment supportive of companion animals. Thus, in a real sense, we view ourselves optimistically and cautiously in a scientific advocacy role for the implementation of such animal-friendly measures, such as a relaxation on the dog prohibition regulation in subsidized housing, and the loosening of restrictions on the accessibility of dogs to public transportation and parks, put into place largely due to concerns about hygiene, safety, and potential conflicts. This small project, despite its professed limitations, along with other initiatives pursued by advocacy groups, has set out to foreground the issue of loneliness and alienation in society [49], and to encourage the general population and its various government and administrative bodies to reconsider their attitudes. There is a general consensus among researchers that given supportive social environments, companion animals can be a powerful means of ameliorating the harmful effects of loneliness and alienation of those who are oppressed by these sinister epidemics. We are conscious of the limitations of our research methodology and the effects of these on the validity of our findings. However, it is a beginning and may well lead to further research with a much greater survey pool and with appropriate weighting given to each section of society. We remain totally committed to the notion that animal companions can improve the quality and happiness of those upon whom the shadows of loneliness and alienation fall and those who genuinely love their animals when they travel their life journeys together.

## Figures and Tables

**Figure 1 ijerph-16-01729-f001:**
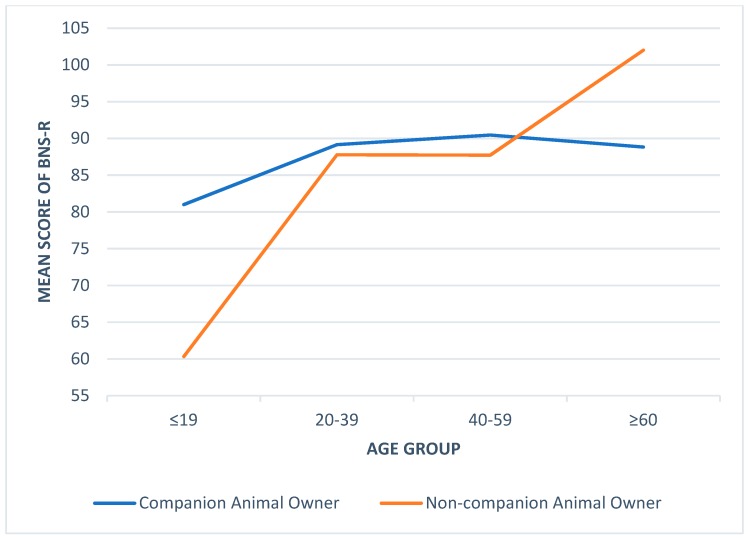
The comparison between owner and non-owner on BNS-R among age groups.

**Figure 2 ijerph-16-01729-f002:**
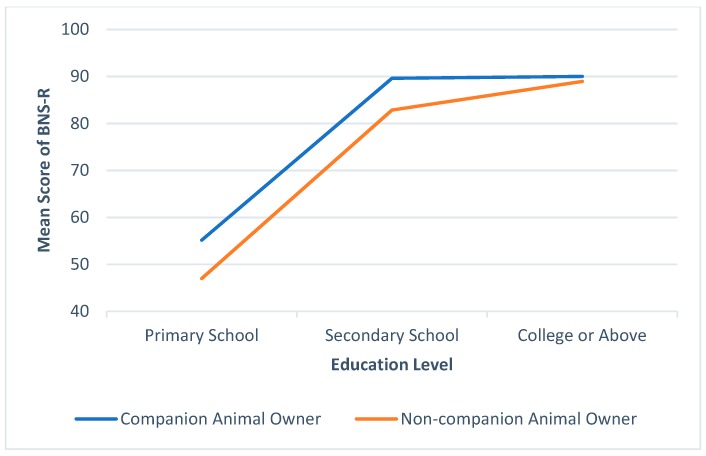
The comparison between owners and non-owners on BNS-R among education levels.

**Table 1 ijerph-16-01729-t001:** Socio-demographic characteristics of owners and non-owners.

Variables	Total (*n* = 986)		Owner (*n* = 674)		Non-Owner (*n* = 312)		χ^2^	*p*
Gender	*n*	%	*n*	%	*n*	%		
Male	197	20	99	15	100	32	39.91	0.00 ***
Female	785	80	575	85	212	68		
Age								
<19	55	6	35	5	20	6		
20–39	573	58	359	53	214	69	25.33	0.00 ***
40–59	341	35	267	40	75	24		
≥60	16	2	13	2	3	1		
Household income (HK$)								
<19,999	228	23	147	22	81	26		
20,000–39,999	321	32	207	31	114	37	8.60	0.01 **
≥40,000	437	44	320	47	117	38		
Education								
Primary or below	13	1	6	1	7	2		
Secondary	307	32	161	24	59	19	12.01	0.00 ***
College or above	639	67	490	73	236	76		
Housing								
Public and subsidised	292	30	164	24	128	41		
Private	567	58	399	59	168	54	42.43	0.00 ***
Village house	127	13	111	17	16	5		
Number of occupants in the household								
1	110	11	90	13	20	6		
2	338	34	263	39	75	24	42.00	0.00 ***
3	233	24	140	21	93	30		
>4	305	31	181	27	124	40		

Statistical significance: * *p* <0.05; ** *p* <0.01; *** *p* <0.001.

**Table 2 ijerph-16-01729-t002:** Descriptive statistics and independent *t*-test results on the physical and psychosocial measures by ownership status.

Variables	Non-Owner			Owner			*t*	*df*	*p*
	*n*	M	SD	*n*	M	SD			
Physical measures									
Number of unhealthy days	238	1.97	4.82	550	2.99	6.57	–2.43	602.40	0.02 **
Number of days walked for 10 mins or more	303	5.00	2.70	659	5.59	2.17	–3.32	487.93	0.00 ***
Psychological measure (DASS 21)									
Depression	312	11.33	4.24	674	11.25	4.35	0.31	984	0.76
Anxiety	312	11.23	3.86	674	10.72	3.87	1.97	984	0.05 *
Stress	312	12.91	4.20	674	12.83	4.41	0.25	984	0.81
Social measure (BNS-R)									
BNS-R	231	86.17	14.71	506	89.46	12.16	–2.97	379.17	0.00 ***

Statistical significance: * *p* < 0.05; ** *p* < 0.01; *** *p* < 0.001. DASS: Depression, Anxiety and Stress Scale. BNS-R: Basic Needs Satisfaction in Relationship Scale.

**Table 3 ijerph-16-01729-t003:** Poisson regression of the impact of age and ownership on the occurrence of unhealthy days.

Variables	Variables		Unhealthy Days			B	SE	Wald Test			Exp(B)
			*n*	M	SD			Wald	*df*	*p*	
Age group	Intercept		-	-	-	0.21	0.25	0.69	1	0.41	1.23
	≥60		13	0.54	1.94	0.64	0.45	1.99	1	0.16	1.90
	40–59		287	2.77	6.29	0.69	0.26	6.79	1	0.01 **	1.96
	20–39		447	2.47	5.59	0.38	0.26	2.22	1	0.14	1.47
	<19		41	5.05	9.64	-	-	-	-	-	1
Age group ownership	≥60	Owner	10	0	0	−30.90	-	-	-	-	-
		Non-owner	3	2.33	4.04						
	40–59	Owner	224	2.84	6.43	0.15	0.09	2.83	1	0.93	1.17
		Non-owner	63	2.52	5.83	-	-	-	-	-	-
	20–39	Owner	288	2.84	6.08	0.45	0.07	43.25	1	0.00 ***	1.57
		Non-owner	159	1.81	4.52	-	-	-	-	-	-
	<19	Owner	28	6.82	11.12	1.71	0.26	43.29	1	0.00 ***	5.54
		Non-owner	13	1.23	2.98	-	-	-	-	-	-

Statistical significance: * *p* < 0.05; ** *p* < 0.01; *** *p* < 0.001.

**Table 4 ijerph-16-01729-t004:** Descriptive statistics on the BNS-R among age group by ownership.

Ownership	Age Group	*n*	M	SD
Yes	≤19	13	81.00	29.85
	20–39	278	89.15	10.73
	40–59	203	90.47	11.93
	≥60	12	88.83	14.19
No	≤19	12	60.33	27.58
	20–39	165	87.78	11.83
	40–59	53	86.72	13.70
	≥60	1	102.00	-

**Table 5 ijerph-16-01729-t005:** ANOVA of impact of ownership and age on BNS-R.

Source	*df*	MS	F	*p*
Household income ^a^	1	2441.32	15.843	0.00 ***
Age group	3	2062.50	13.39	0.00 ***
Ownership	1	141.40	0.92	0.34
Age group * ownership interaction	3	658.69	4.28	0.01 **
Within groups	729	154.10		
Total	737			

^a^ Control variable. Statistical significance: * *p* <0.05; ** *p* <0.01; *** *p* <0.001.

**Table 6 ijerph-16-01729-t006:** Descriptive statistics on BNS-R among education level by ownership status.

Ownership Status	Education Level	*n*	M	SD
Yes	Primary school	6	55.17	29.07
	Secondary school	180	89.61	10.98
	College or above	320	90.03	11.44
No	Primary school	7	47.00	29.66
	Secondary school	57	82.86	14.12
	College or above	167	88.95	11.16

**Table 7 ijerph-16-01729-t007:** ANOVA of impact of ownership and education level on BNS-R.

Source	*df*	*MS*	*F*	*p*
Household income ^a^	1	1664.06	12.08	0.00 ***
Education level	2	8008.95	56.93	0.00 ***
Ownership	1	587.31	4.175	0.04
Education level * ownership	1	536.26	3.81	0.02 *
Within groups	730			
Total	736			

^a^ Control variable. Statistical significance: * *p* < 0.05; ** *p* < 0.01; *** *p* < 0.001.

**Table 8 ijerph-16-01729-t008:** Comparison of physical, psychological and social well-being between dog owners and non-dog owners.

Variables	Dog Owner			Non-Dog Owner			*t* Statistic		
	*n*	M	SD	*n*	M	SD	*t*	*df*	*p*
Physical well-being									
Unhealthy days	397	2.77	6.05	150	3.63	7.81	−1.22	220.04	0.22
10 min walk	472	5.72	2.07	183	5.29	2.32	2.2	300.71	0.29
Psychological well-being									
Depression	483	11.27	4.59	187	11.15	3.47	0.36	444.60	0.72
Anxiety	483	10.67	4.00	187	10.77	3.30	−0.31	407.22	0.75
Stress	483	12.81	4.52	187	12.86	4.01	−0.16	378.51	0.88
Social well-being									
Autonomy	385	14.08	2.22	138	14.19	1.51	−0.65	356.07	0.52
Competence	376	12.94	2.42	138	13.08	1.61	−0.71	365.91	0.48
Relatedness	384	17.46	3.40	142	17.60	2.86	−0.45	296.84	0.65

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
