# Peer review of "Companion Animal Ownership and Human Well-Being in a Metropolis—The Case of Hong Kong"

_ijerph, 2019, doi:10.3390/ijerph16101729_

Reviewer 1 Report

There are several problems with this text which have made me to suggest rejection of the paper.

Major concerns are based on the description of the statistics and the presentation of the statistics. For example, there is no reference to statistical software, figures can't be understood due to lack of legend and tables could be much more clearer so that the star-system may be avoided and real p-values displayed.

Parts of methods appear in the results section and parts of discussion appear in the conclusion. I also find the link between the discussion and the conclusion weak.

There is also a vagueness in what ownership means. When is it referring to pet animals and when to dogs.

I also question the use of single items from a questionnaire and give them that much explanation value for something as complex as physical health and experiences of physical health. 

Author Response

Reviewer 1.

R: There are several problems with this text which have made me to suggest rejection of the paper.

R: Major concerns are based on the description of the statistics and the presentation of the statistics. For example, there is no reference to statistical software, figures can't be understood due to lack of legend and tables could be much more clearer so that the star-system may be avoided and real p-values displayed.

A: We thank the reviewer for this and we have edited and adjusted the results section extensively.

R: Parts of methods appear in the results section and parts of discussion appear in the conclusion. I also find the link between the discussion and the conclusion weak.

A: We thank the reviewer for this and we have edited and adjusted the results section extensively.

R: There is also a vagueness in what ownership means. When is it referring to pet animals and when to dogs.

A: We have now defined very clearly what we meant by companion animal ownership, dog owners, non-dog owners, and non-owners throughout the manuscript.

R: I also question the use of single items from a questionnaire and give them that much explanation value for something as complex as physical health and experiences of physical health. 

A: We also acknowledged this might be a significant limitation of the study. However, after careful consideration at the measurement development and analysis stages, we decided that this single item might be the most appropriate item to represent a less subject opinion of participants’ physical well-being. Having said that, we also acknowledged that this in our limitation section.

Reviewer 2 Report

Overall I think paper could provide a contribution to the existing field of research. However, further explanations are needed throughout. It is important that paper is check for spelling and readability. 

Introduction

In general attention needs to be paid to correct use of English spelling and grammar. Although the introduction is succinct I feel it is lacking a more comprehensive rationale for the study. Some consideration to other literatures which have found effects between socio-demographic factors and wellbeing of companion animal owners would be useful – e.g. there are a number of studies that could be used to reference the relationship between wellbeing and owner age (impact in younger and older population), owner gender and neuro-developmental status. I am not suggesting a detailed review but this is lacking. The authors should also consider that in some populations no beneficial effect has been found – reference to this is lacking.   

Line 32: diminishing instrumental value – due to increased technologies? Make this point clear.

Line 32-33: ‘and the increased of anthropomorphism in urbanized societies’ – this does not make sense, plus the point you are trying to make is not clear.

Line 40: Owning not owing – psychical – do you mean physical?

Line 46-47: Although walking the dog and ‘health enhancing activities’ may partly under lie the mechanistic process, this is still very much up for debate and should not be written so conclusively. There are also other incidental and psychological processes that may not actually be ‘activities’.

Methods

Line 102: References to the Health QOL scale used are needed as I am unclear on the specific scale items employed here. It is also unclear as to why one single question was taken?

When was data collected from (time period)?

Results

Avoid repeating data in text that is in the tables.

Table 1 – check consistency and formatting (e.g. N is sometimes large and sometimes small)

Table 2 – it is standard to report SD or SE – not both.

Line 181 – statically significant lower – not statistical  

Line 183 – use the term ‘not statistically significant’ rather than insignificant

My concern here is that you group owners vs non-owners, but the different type of companion animal owned is likely to have a big impact – e.g. they are unlikely to walk a turtle.

Later on you look at just dog owners vs non-dog owners – what proportion of your sample is this? It may have been useful to explore the pet-owner relationship more, e.g relationships between attachment, or length of ownership, time spent with animal. This should be included in the discussion. Simply owning a pet does not necessarily lead to benefits any more than owning a running machine does – it depends how often and how long you use it for!

Discussion

The information on who were pet owners was useful and interesting. Findings on age status need relating to previous literature. In general there is a lack of real discussion about the results, what these mean and how these compare to previous literatures.

Line 289 – Larger? Not laxer

Author Response

Reviewer 2:

Overall I think paper could provide a contribution to the existing field of research. However, further explanations are needed throughout. It is important that paper is check for spelling and readability. 

A: Thank you, we had it proof-read and edited extensively by a native writer.

Introduction

In general attention needs to be paid to correct use of English spelling and grammar. Although the introduction is succinct I feel it is lacking a more comprehensive rationale for the study. Some consideration to other literatures which have found effects between socio-demographic factors and wellbeing of companion animal owners would be useful – e.g. there are a number of studies that could be used to reference the relationship between wellbeing and owner age (impact in younger and older population), owner gender and neuro-developmental status. I am not suggesting a detailed review but this is lacking. The authors should also consider that in some populations no beneficial effect has been found – reference to this is lacking.   

A: Thank you and we have expand the rationale of the study by citing that loneliness is prevalent in urban areas, studying human-animal interactions has become an emerging research trend, and more mixed results on ownership and well-being are being identified lately.

Line 32: diminishing instrumental value – due to increased technologies? Make this point clear.

A: Revised, and thank you.

Line 32-33: ‘and the increased of anthropomorphism in urbanized societies’ – this does not make sense, plus the point you are trying to make is not clear.

A; Revised, and thank you.

Line 40: Owning not owing – psychical – do you mean physical?

A; Revised, and thank you.

Line 46-47: Although walking the dog and ‘health enhancing activities’ may partly under lie the mechanistic process, this is still very much up for debate and should not be written so conclusively. There are also other incidental and psychological processes that may not actually be ‘activities’.  –

A; Revised, and thank you for bringing this up. We have made our writings not so much conclusively.

 Methods

 Line 102: References to the Health QOL scale used are needed (ref 33) as I am unclear on the specific scale items employed here. It is also unclear as to why one single question was taken? 

A: We also acknowledged this might be a significant limitation of the study. However, after careful consideration at the measurement development and analysis stages, we decided that this single item might be the most appropriate item to represent a less subject opinion of participants’ physical well-being. Having said that, we also acknowledged that this in our limitation section.

When was data collected from (time period)?

A; Thank you and we added this information in the methods section.

 Results

 Avoid repeating data in text that is in the tables.

A: Thank you for pointing this out, and we have tried to avoid doing that in the revised version,

Table 1 – check consistency and formatting (e.g. N is sometimes large and sometimes small)

A: Thank you. The N represents the overall number of people, and the n represents the subgroups. We have checked carefully and should have avoided presenting a confusing message.

Table 2 – it is standard to report SD or SE – not both.

A; Thank you, and we have adjusted this in the revised version.

 Line 181 – statically significant lower – not statistical  

A: Thank you and we have changed all of these wordings in the revised version.

Line 183 – use the term ‘not statistically significant’ rather than insignificant (Check and adjusted)

A: Thank you and we have changed all of these wordings in the revised version.

My concern here is that you group owners vs non-owners, but the different type of companion animal owned is likely to have a big impact – e.g. they are unlikely to walk a turtle. –

A: Thank you for raising this up and that was the rationale of our 3rd analysis that was to see whether owing dogs would be different than those who won animals other than dogs. We found no statistically significant findings.

Later on you look at just dog owners vs non-dog owners – what proportion of your sample is this?

A; Thank you and we have elaborated on that.

Discussion

The information on who were pet owners was useful and interesting. Findings on age status need relating to previous literature. In general there is a lack of real discussion about the results, what these mean and how these compare to previous literatures.

A: Thank you. Given that our study was exploratory and we did not find a lot of statistically significant findings, to minimize over interpretation, we limited our discussion at the exploratory level.

Line 289 – Larger? Not laxer

A: Thank you, and this was corrected.

Reviewer 3 Report

The current manuscript is based on a statistical analysis which is not the reviewer’s expertise; therefore, the statistical analysis part was not reviewed expertly.

By conducting quantitative research, this manuscript aims to examine the influence of companion animal ownership on human well-being in an over-urbanized environment. The analysis results show that socio-demographic factors have stronger impacts on human well-being compared with companion animal ownership. As discussed by the authors, this finding is different from previous research in this field. The results can provide policy implications in an over-urbanized metropolis. Guided by the concept of One Health, this research proposes that there is a need to build a more animal-friendly environment which can benefit human, animal, and environmental health. There are some comments and suggestions.

1. On page 4, table 1 is about “socio-demographic characteristics of owners and non-owners”. In the part of household income, there are 498 people with a household income of 20,000-39,999 (51%), but in this group there are 207 owners and 114 non-owners which is 314 in total. Also, as shown in this table, there are 260 people with a household income over 40,000 (26%), and in this group there are 320 owners and 117 non-owners which is 437 in total. The numbers in the two groups do not match. Based on these numbers, the majority of the participants were with a household income of over 40,000, not 20,000-39,999 as shown in line 144 in this manuscript.

A related question is that the paragraph starting from line 149 talks about the T-test results of “the differences in the socio-demographic characteristics among owners and non-owners”. Is this a T-test or cross-tabulation with Chi-square analysis?

2. The contents from line 56 to line 63 talk about the “unfriendly environment of co-habitation with companion animals”. In the paragraph starting from line 64, the authors describe the concept of One Health which concentrates on “the interdependence of human health, animal health, and the health of the ecosystem”. Is there any evidence showing how this animal-unfriendly environment affects the health of companion animals and animal welfare?

3. The contents from line 323 to line 325 mention that “the finding that companion animal ownership with better social well-being with their significant others who are more likely to live together may suggest that animal ownership acts as social lubricants between known people but not neighbours in Hong Kong”. The social well-being is this research is measured by the Basic Needs Satisfaction in Relationship Scale (page 3, paragraph starting from line 115). This scale is about the “experience when relating with a significant other” (line 120-121) and the “significant other was defined as a spouse and/or best friend” (line 122-123).

The social well-being mainly focuses on significant others, and this may not well support the conclusion that “animal ownership acts as social lubricants between known people but not neighbours in Hong Kong”.

4. The introduction of One Health to the paper is very limited. I woudl suggets removing it or fully expanding upon it.

5. Line 58 – what does this mean “an unfriendly environment of co-habitation with companion anaimls”.

6. Line 74, research focus #4, this is not indicated in the abstract. And I am not clear where it is in the findings.

7. Line 296 – “resonates with...” – is this correct?

8. Line 354 – What is means by the diminsihing utilitarian role of domestic animals?

Author Response

Reviewer 3:

The current manuscript is based on a statistical analysis which is not the reviewer’s expertise; therefore, the statistical analysis part was not reviewed expertly.

By conducting quantitative research, this manuscript aims to examine the influence of companion animal ownership on human well-being in an over-urbanized environment. The analysis results show that socio-demographic factors have stronger impacts on human well-being compared with companion animal ownership. As discussed by the authors, this finding is different from previous research in this field. The results can provide policy implications in an over-urbanized metropolis. Guided by the concept of One Health, this research proposes that there is a need to build a more animal-friendly environment which can benefit human, animal, and environmental health. There are some comments and suggestions. –

A; Thank you and we agreed with the reviewer’s comment. Hence, we have deleted all the discussions about One Health in the revised manuscript.

1. On page 4, table 1 is about “socio-demographic characteristics of owners and non-owners”. In the part of household income, there are 498 people with a household income of 20,000-39,999 (51%), but in this group there are 207 owners and 114 non-owners which is 314 in total. Also, as shown in this table, there are 260 people with a household income over 40,000 (26%), and in this group there are 320 owners and 117 non-owners which is 437 in total. The numbers in the two groups do not match. Based on these numbers, the majority of the participants were with a household income of over 40,000, not 20,000-39,999 as shown in line 144 in this manuscript. –

A: Thank you for this careful observation. We have corrected it.

A related question is that the paragraph starting from line 149 talks about the T-test results of “the differences in the socio-demographic characteristics among owners and non-owners”. Is this a T-test or cross-tabulation with Chi-square analysis?

A: Thank you for this careful observation. We have corrected it.

2. The contents from line 56 to line 63 talk about the “unfriendly environment of co-habitation with companion animals”. In the paragraph starting from line 64, the authors describe the concept of One Health which concentrates on “the interdependence of human health, animal health, and the health of the ecosystem”. Is there any evidence showing how this animal-unfriendly environment affects the health of companion animals and animal welfare?

A: Thank you and we agreed with the reviewer’s comment. Hence, we have deleted all the discussions about One Health in the revised manuscript.

3. The contents from line 323 to line 325 mention that “the finding that companion animal ownership with better social well-being with their significant others who are more likely to live together may suggest that animal ownership acts as social lubricants between known people but not neighbours in Hong Kong”. The social well-being is this research is measured by the Basic Needs Satisfaction in Relationship Scale (page 3, paragraph starting from line 115). This scale is about the “experience when relating with a significant other” (line 120-121) and the “significant other was defined as a spouse and/or best friend” (line 122-123).

A: Thank you and we have revised our writings and discussion and suggesting that we only studied the impacts with significant others, but not other relationships.

The social well-being mainly focuses on significant others, and this may not well support the conclusion that “animal ownership acts as social lubricants between known people but not neighbours in Hong Kong”.

A: Thank you and we have revised our writings and discussion and suggesting that we only studied the impacts with significant others, but not other relationships.

4. The introduction of One Health to the paper is very limited. I would suggest removing it or fully expanding upon it.

A: Thank you and we agreed with the reviewer’s comment. Hence, we have deleted all the discussions about One Health in the revised manuscript.

5. Line 58 – what does this mean “an unfriendly environment of co-habitation with companion animals”.-

A; We deleted sentences like this in the revised version to minimise confusions. Thank you.

6. Line 74, research focus #4, this is not indicated in the abstract. And I am not clear where it is in the findings.

A; Thank you and we deleted it.

7. Line 296 – “resonates with...” – is this correct?

A: We used echos instead of resonates. Thank you.

8. Line 354 – What is means by the diminishing utilitarian role of domestic animals?

A: This is deleted. Thank you.

Response to reviewer

I don't think the statistical methods are entirely appropriate in this research and I think that the findings of any benefit of animal ownership are very weak. Here are some specific comments.

1. The models are not correctly specified.

The model for unhealthy days should probably use a Poisson regression (not linear) and it's not clear what is hte best model for the other outcomes

A; Thank you. Poission regression was used in the revised version.

2. Age is not correctly handled

Age should not be entered into the model as a continuous variable.

A: Age is now entered as a categorical variable.

3. They should properly handle income

In HK I would expect that pet ownership is heavily determined by wealth. This needs to be handled in the model.

A; Thank you and we have now included income level as a significant factor in the analysis.

4. The models do not explain much

The R-squared values of these models are too low to justify publication (they explain almost no variation in the outcomes)

A: No R-squared values are now being reported and we have also stated a few times that given the limitations of our study, interpretation of findings must be done in caution.

Authors should make it a more descriptive article just describing the lack of relationship between companion animals and health in Hong Kong.

A; THANK YOU and we made the manuscript very descriptive.

Round  2

Reviewer 1 Report

Line 16 - six hundred and four should be 674.

Line 100 - omit 20 in "19th July 20 2016"

Line 119 - Should it be "Physical well-being" instead of "physical health"

However, I still find the statistical analysis section in the methods a little too short so that choices made in the poisson is hard to follow.

Author Response

We sincerely thank the review for his/her careful observations. 

Line 16 - six hundred and four should be 674. - Thank you. Changed.

Line 100 - omit 20 in "19th July 20 2016" - Thank you. Changed.

Line 119 - Should it be "Physical well-being" instead of "physical health" - Thank you. Changed. 

However, I still find the statistical analysis section in the methods a little too short so that choices made in the Poisson is hard to follow. - Thank you and we expanded a little more about the Possion to line 200-204: 

'To explore the moderation effect of companion animal ownerships between sociodemographic and well-being variables, Poisson Regression was conducted for physical well-being as the response variable that is a continuous variable. Physical well-being was represented as the number of unhealthy days in the past month. ANOVA was conducted for psychosocial well-being variables as they are categorical variables. '

Once again, we sincerely thank the Editor and all reviewers for their invaluable comments and suggestions. 
